# Anti-Coronavirus Activity of Chitosan-Stabilized Liposomal Nanocarriers Loaded with Natural Extracts from Bulgarian Flora

**DOI:** 10.3390/life14091180

**Published:** 2024-09-19

**Authors:** Anna Gyurova, Viktoria Milkova, Ivan Iliev, Nevena Lazarova-Zdravkova, Viktor Rashev, Lora Simeonova, Neli Vilhelmova-Ilieva

**Affiliations:** 1Institute of Physical Chemistry ‘Acad. R. Kaischew’, Bulgarian Academy of Sciences, 1113 Sofia, Bulgaria; any_gyurova@abv.bg (A.G.); vmilkova@ipc.bas.bg (V.M.); 2Institute of Experimental Morphology, Pathology and Anthropology with Museum, Bulgarian Academy of Sciences, 1113 Sofia, Bulgaria; taparsky@abv.bg; 3Department of Biotechnology, University of Chemical Technology and Metallurgy, 8 Kliment Ohridski, 1756 Sofia, Bulgaria; nevena@uctm.edu; 4Department of Virology, Stephan Angeloff Institute of Microbiology, Bulgarian Academy of Sciences, 26 Georgi Bonchev, 1113 Sofia, Bulgaria; vpr2012@abv.bg

**Keywords:** natural extracts, chitosan, liposomes, encapsulation, drug release, coronavirus HCoV-OC43, antiviral activity, natural inhibitors of viral replication, cytotoxicity, phototoxicity

## Abstract

Disease’s severity, mortality rates, and common failures to achieve clinical improvement during the unprecedented COVID-19 pandemic exposed the emergency need for new antiviral therapeutics with higher efficacy and fewer adverse effects. This study explores the potential to encapsulate multi-component plant extracts in liposomes as optimized delivery systems and to verify if they exert inhibitory effects against human seasonal betacoronavirus OC43 (HCoV-OC43) in vitro. The selection of *Sambucus nigra*, *Potentilla reptans*, *Allium sativum*, *Aesculus hippocastanum*, and *Glycyrrhiza glabra* L. plant extracts was based on their established pharmacological and antiviral properties. The physicochemical characterization of extract-loaded liposomes was conducted by DLS and electrokinetics. Encapsulated amounts of the extract were evaluated based on the total flavonoid content (TFC) and total polyphenol content (TPC) by colorimetric methods. The BALB 3T3 neutral red uptake (NRU) phototoxicity/cytotoxicity assay was used to estimate compounds’ safety. Photo irritation factors (PIFs) of the liposomes containing extracts were <2 which assigned them as non-phototoxic substances. The antiviral capacities of liposomes containing medicinal plant extracts against HCoV-OC43 were measured by the cytopathic effect inhibition test in susceptible HCT-8 cells. The antiviral activity increased by several times compared to “naked” extracts’ activity reported previously. *A. hippocastanum* extract showed 16 times higher inhibitory properties reaching a selectivity index (SI) of 58.96. Virucidal and virus-adsorption effects were investigated using the endpoint dilution method and ∆lgs comparison with infected and untreated controls. The results confirmed that nanoparticles do not directly affect the viral surface or cell membrane, but only serve as carriers of the active substances and the observed protection is due solely to the intracellular action of the extracts.

## 1. Introduction

Natural extracts of botanical origin have been used in traditional medicine as cures for various malaises throughout all human civilization. By date, it is established that they contain diverse biologically active ingredients, such as flavonoids, saponins, terpenes, essential oils, alkaloids, etc. [1,2]. Herbal products tend to become a preferred valuable adjuvant and/or an alternative to conventional synthetic drugs, having certain advantages such as a relatively higher biocompatibility, minimal side effects, good safety profile allowing prolonged prophylactic administration, etc. Numerous medicinal plants have been found to possess antitumor [3,4], antibacterial [5,6], antiviral [7,8], immunomodulatory [9], antidiabetic [10], anticholesterolemic [11], antiulcerolitic [12], anti-inflammatory [13], and antioxidant [13,14] activities. Additionally, floral preparations are also widely used in food and cosmetic industries.

To date, there are no effective specific therapeutics approved for coronavirus infections, including severe acute respiratory syndrome coronavirus-2 (SARS-CoV-2) treatment. Some synthetic therapeutics have been developed which relieve symptoms and shorten the recovery time to some extent, but these are usually accompanied by numerous side effects. Additionally, they have been found to promote the emergence of drug resistance among circulating viruses, leading to therapeutic clinical failure. Therefore, new resources, preferably of natural origin, should be exploited since they cause fewer adverse events to the patient, have better bioavailability, and rarely result in the selection of resistant mutants.

Currently, extensive research is being conducted on the antiviral, anti-inflammatory, and antioxidant capacities of botanical products as potential novel medical approaches [15,16,17]. Our work reveals a mechanism for increasing the antiviral potential of five medicinal plant extracts that are widely distributed in the Bulgarian natural environment. The selection of the extracts was based on our previous results [18,19], in which a large group of medicinal herbal preparations were investigated for their antiviral activity against human coronavirus (HCoV-OC43 and HCoV-229E), herpes simplex virus type 1, human adenovirus (HAdV-5), and poliovirus 1. Those that showed the highest anti-coronavirus activity were included in the present study, namely *Sambucus nigra* (elderberry) [20,21,22,23], *Potentilla reptans* (creeping cinquefoil) [24,25], *Allium sativum* (garlic) [26,27], *Aesculus hippocastanum* (horse chestnut) [28,29], and *Glycyrrhiza glabra* L. (licorice) [30,31,32]. The plant part used for the production of the extract, the period of collection of the plant material, the main components contained in the extract, as well as some of the biological activities of the studied extracts described so far are presented in Table 1. Along with the numerous advantages of phytoextracts and polyphenols as their important ingredients, several drawbacks have been reported, such as low stability against environmental conditions (oxidation, temperature, pH, ionic strength, etc.), poor bioavailability (low intestinal adsorption), limited solubility in water, interaction with proteins resulting in aggregation and polyphenol losses, worsened organoleptic properties, etc. [2,33]. To solve the downsides described, for medical, dietary, and other applications, herbal materials are preliminarily encapsulated into suitable carriers, such as polymeric nanoparticles, cyclodextrins, micelles, vesicles, micro and nanoemulsions, etc. [1,2,34,35,36,37], thus preserving the active components and ensuring safe delivery to the affected tissues targeted to be released.

Liposomes represent a version of vesicles, where the bilayer(s) is/are constructed by phospholipids that are commonly present in the natural biomembranes (non-toxic, biocompatible, non-immunogenic) and possess significantly better bioadhesive properties to the target cells compared to the loaded drugs/extracts alone. Among their content-protective functions is the ability to survive in the gastric environment and securely reach the intestinal tract which promotes absorption rates (enhanced bioavailability) [2,37]. Additionally, this type of carrier is also an appropriate vehicle for polyphenols/flavonoids due to their amphiphilic properties, meaning they are capable of packing hydrophilic, hydrophobic, and surfactant-like compounds either into the aquatic core or incorporating (partially or completely) them into the different parts of the bilayer. Acidic conditions of pH 3.8 are recommended for the encapsulation of polyphenols into liposomes to avoid the oxidation of extracts [38,39,40].

**Table 1 life-14-01180-t001:** The plant part used for the production of the extract, the period of collection of the plant material, the main components contained in the extract, and the biological activities of the studied extracts.

Plant Species		Part Used	Raw Material Collection Period	Main Components of the Composition	Biological Activities	References
*Aesculus hippocastanum* (horse chestnut)	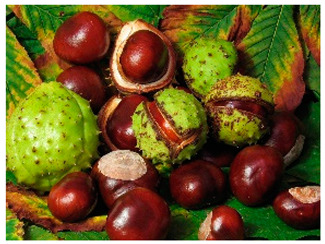	Seed	Mid-autumn	saponins coumarins polyphenols flavonoids	Anti-inflammatory, vasoprotective, immunomodulatory, antioxidant, virucidal, anti-RSV, HSV-1, VSV, RSV, Dengue virus effects.	[41,42]
*Allium sativum* (garlic)	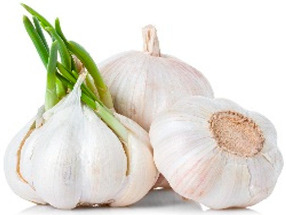	Bulb	Summer	alliin sulfur compounds alkaloids tannins flavonoids	Immunomodulatory activity; prevention of infectious diseases; pronounced antiviral activity through various mechanisms of action: inhibition of virus entry into the cell, inhibition of viral RNA polymerase, reverse transcriptase, DNA synthesis.	[7]
*Sambucus nigra* (elderberry)	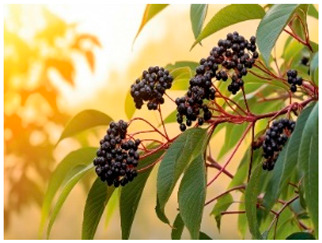	Fruit	Late summer–autumn	triterpenoids polyphenols flavonoids anthocyanidins proanthocyanidins	Anti-inflammatory, immunomodulatory, antiviral activity.	[8,23]
*Glycyrrhiza glabra* L. (licorice)	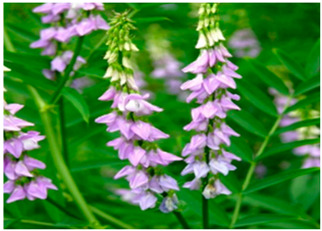	Root	Late autumn or early spring	glycyrrhizin polyphenols flavonoids	Positive effects in gastrointestinal problems (gastritis, peptic ulcer), in respiratory infections, arthritis, and tremors. Marked anti-inflammatory, antispasmodic, antioxidant, antidiabetic, antimalarial, antimycotic, antibacterial, antiviral effects.	[43,44,45]
*Potentilla reptans* (creeping cinquefoil)	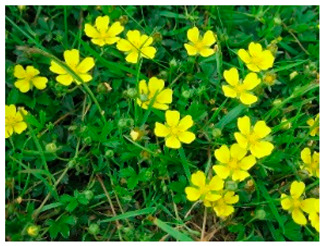	Stem	Late spring and summer	saponins polyphenols flavonoids catechins	Pronounced antidiarrheal, antidiabetic, hepatoprotective, antioxidant, antispasmodic, anti-inflammatory, antitumor, antifungal, antibacterial, antiviral actions.	[46]

However, the insufficiency of simple (naked) liposomes is based on their kinetic instability which leads to leakage of the loaded content over time. To fix the problem, their proper design requires additional coating by polymers, such as biocompatible polysaccharides [40,47,48,49,50]. Hereby, chitosan is selected to envelope the liposomal carriers because of its well-established properties for the preparation of drug delivery systems [51,52]. A great variety of chitosans with various molecular weights and degrees of acetylation exist as the latter being strongly related to the electric charge possessed. The polymer interacts mainly with the lipids, governed predominantly by the electrostatic attraction of chitosan NH_3_^+^ to the negatively charged lipid heads, and to a lesser extent by H-linking between chitosan H_2_N and lipid OH, as well as hydrophobic forces, which could cause partial anchoring of the polysaccharide’s hydrophobic portions into the acyl chain space of the bilayer [48].

The purpose of study is to investigate stabilized liposomal carriers as potential encapsulating agents of multi-component plant extracts from Bulgarian medicinal plants *Sambucus nigra*, *Potentilla reptans*, *Allium sativum*, *Aesculus hippocastanum*, and *Glycyrrhiza glabra* L. (Table 1). The combinations of phospholipid 1,2-dioleoyl-sn-glicero-3-phoshocholine (DOPC) and three types of chitosans with various molecular weights and degrees of acetylation, adsorbed in different concentrations, are studied as variants for encapsulating and protective vehicles of potential coronavirus inhibitors.

## 2. Materials and Methods

### 2.1. Materials

#### 2.1.1. Researched Products

##### Plant Extracts

Dry aqueous ethanol (15% ethanol) extracts of five Bulgarian medicinal plants (*Sambucus nigra*, *Allium sativum*, *Potentilla reptans*, *Aesculas hippocastanum*, and *Glycyrriza glabra* L. (Table 1)) were prepared and provided by Extractpharma Ltd., Sofia, Bulgaria, as described previously [18,19]. In brief, the procedure included an 18 h extraction under atmospheric pressure and 40 °C temperature. The yields of plant essences were concentrated and dried in a vacuum-drying or powder-drying device depending on the type of herb, to obtain the dry material. The stock solutions of plant extracts were prepared in a concentration of 10 mg/mL in bidistilled water. The solutions were filtered through a 5 µm filter (filtraTECH, Saint Jean de Braye, France) to remove the insoluble components. The produced colored and clear solutions were encapsulated into the liposomes.

##### Reference Substance

The stock solution of Veklury^®^ (Gilead Science Inc. Ireland UC, IDA Business & Technology Park, County Cork, Ireland) with a concentration of 150 mg/mL was prepared in bidistilled water and the concentration of remdesivir (REM) in the stock solution was estimated as 8.3 × 10^−3^ M.

#### 2.1.2. Polysaccharides and Lipids

Chitosans, CS, (product numbers 448869, 448877, 523682) were purchased from Sigma Aldrich: CS-L (Mw 50–190 kDa, DA 75–58%), CS-H (Mw 190–310 kDa, DA 75–58%), and COS (Mw 5 kDa, DA < 10%). The stocks were dissolved in hydrochloric acid at a concentration of 1 mg/mL with pH 4. Chitosan oligosaccharide and COS were diluted in bidistilled water. Prior to the utilization, the solutions were filtered through a 0.45 µm filter (Minisart^®^, Sartorius, Göttingen, Germany) to discard potential aggregates. 1,2-dioleoyl-sn-glicero-3-phoshocholine (DOPC, chloroform solution, 25 mg/mL) phospholipid product (Avanti Polar Lipids Inc., Birmingham, UK) served to produce unilamellar liposomes.

#### 2.1.3. Light Source

The illumination was performed by an artificial solar light simulator Helios-iO, model LE-9ND55-H—5500K (SERIC Ltd., Tokyo, Japan).

#### 2.1.4. Reagents

Cell cultivation: DMEM (Dulbecco’s modified Eagle’s medium) with high (4.5 g/L) glucose, RPMI 1640 (Roswell Park Memorial Institute Medium), fetal bovine serum (FBS), horse serum (HS), antibiotics (penicillin and streptomycin), L-glutamine, and neutral red were supplied by Sigma-Aldrich, Schnelldorf, Germany. The disposable consumables were manufactured by Orange Scientific, Braine-l’Alleud, Belgium.

#### 2.1.5. Cells and Virus

Mouse embryonic fibroblasts (BALB/3T3 clone A31) and human colon carcinoma (HCT-8) cell lines were obtained from American Type Cultures Collection (ATCC, Manassas, VA, USA). BALB/3T3 cells were cultured in 75 cm^2^ tissue culture flasks in DMEM, 10% FBS, and antibiotics (penicillin 100 U/mL and streptomycin 100 µg/mL) at 37 °C, 5% CO_2_, and 90% humidity. Permanent HCT-8 cells were maintained at 37 °C and 5% CO_2_ using sterile RPMI 1640 medium containing 0.3 mg/mL L-glutamine, 10% horse serum, 100 UI penicillin, and 0.1 mg streptomycin/mL.

The human coronavirus OC-43 (HcoV-OC43, ATCC: VR-1558) strain was propagated in HCT-8 in RPMI 1640 medium with 2% horse serum, 100 U/mL penicillin, and 100 μg/mL streptomycin added. Cells were destroyed 5 days after the infection initial timepoint by double freezing and thawing as the virus was further titrated according to the endpoint dilution assay [53]. Virus and mock aliquots were stored at −80 °C.

### 2.2. Methods

#### 2.2.1. Formation of the Liposomes

Liposomes were produced by the thin-film hydration method. A total of 200 µL of lipid in chloroform solution (25 mg/mL) was dried under a stream of nitrogen by rotation to form a thin lipid layer on the flask wall. To obtain unloaded or extract-loaded liposomes (Figure 1), it was placed in 2 mL bidistilled water or in an extract solution (10 mg/mL) to a final lipid concentration of 2.5 mg/mL. The solution was frozen in liquid nitrogen by four consecutive freezing/heating cycles and the stock was subjected to ultrasound for 15 min in an ice bath. To avoid possible conglomeration during the subsequent steps, the lipid content of the dispersion was adjusted to 0.02 mg/mL with HCl at pH 4 followed by extrusion through a 0.20 µm filter (Minisart^®^, Sartorius). The liposome concentration in the samples was estimated as 3.5 × 10^14^ particles/mL. To optimize the loaded liposomes’ stability, they were covered with a layer of chitosan. The procedure included an addition of 2 mL of diluted dispersion of liposomes to the solution of positively charged chitosan (with the required concentration) and stirring for 20 min.

#### 2.2.2. Quantification of Encapsulated Extracts

Because of the multi-component composition of the extracts, the loaded amount was determined toward the total water-soluble flavonoid or polyphenol content. The procedure for the determination of the encapsulated amount was implemented as follows: the stock aqueous dispersion of extract-carrying liposomes was centrifuged at 15,000 rpm at (21,382× *g*, 15 °C) for 90 min and the supernatant was extracted. The total flavonoid content (TFC) was estimated and expressed in quercetin equivalents using the colorimetric assay described by Gouveia and Castiho [54]. A mixture of 0.30 mL methanol, 0.02 mL of 10 wt% solution of AlCl_3_, and 0.56 mL double distilled water was combined with 0.50 mL supernatant of the dispersion of liposomes loaded with extract. The vials were incubated at 25 °C in an ES-20 (Biosan SIA, Riga, Latvia). TFC was estimated as micrograms of quercetin by monitoring with a T60 UV–visible spectrophotometer (PG Instruments Limited, Woodway Lane, Alma park, Leicestershire, UK). The compound was detected at a wavelength of 415 nm corresponding to the maximum absorbance peak. For the determination of the loaded amount of TFC, the calibration curve of quercetin was used. It was obtained by using the same procedure, but the mixture (from methanol, solution of AlCl_3_ and water) was combined with 0.50 mL of quercetin solution in methanol concentrations in the range of 10^−4^–2 mg/mL.

The total polyphenol content (TPC) was estimated and expressed in gallic acid equivalents by using the Folin–Ciocalteu assay [55]. To the 0.15 mL supernatant of the dispersion of liposomes loaded with extract in an Eppendorf tube was added 0.75 mL Folin–Ciocalteu’s reagent (diluted 1:10 with double distilled water) and 0.6 mL Na_2_CO_3_ (7.5 wt%). The tubes were incubated in an incubator for 10 min at 50 °C. TPC was estimated as micrograms of gallic acid. The absorbance was measured at 760 nm by monitoring with a spectrophotometer. The calibration curve of gallic acid was obtained by the same procedure, but the supernatant was replaced with a solution of gallic acid of concentrations ranging from 10^−4^ to 3 mg/mL.

To evaluate the encapsulation efficiency, the TFC and TPC values (expressed in quercetin or gallic acid equivalents) in the initial solution of the extracts were determined using the same procedures, but the supernatant was replaced with aqueous solutions of pure extracts (10 mg/mL).

The encapsulation efficiency (EE%) of TFC and TPC in the liposomes was calculated based on quercetin or gallic acid equivalents by using the relation
(1)$$\frac{\left(C_{extract}^{equivalents}-C_{free}^{equivalents}\right)}{C_{extract}^{equivalents}}\times 100\%$$
where $C_{extract}^{equivalents}$ is the estimated TFC and TPC content in the stock extract solution expressed in quercetin or gallic acid equivalents, respectively, and $C_{free}^{equivalents}$ is the estimated content of the components in the supernatant after the encapsulation.

#### 2.2.3. Liposomes’ Electrokinetic Charge and Size Determination

The liposomes’ hydrodynamic diameter was measured by dynamic light scattering with non-invasive backscattering (DLS-NIBS, measuring angle 173°). The ζ-potential was determined by mixed-mode measurements of phase analysis light scattering. Zetasizer Pro (Malvern Panalytical Ltd., Malvern, UK) equipped with a He-Ne laser with a maximum power of 10 mW operating at a wavelength of 633 nm and a fixed scattering light angle of 173° was used for light scattering experiments. All measurements were performed at 24.0 ± 0.1 °C. The mean of five measurements was taken for the capsules’ electrokinetic potential and size estimation.

#### 2.2.4. Release of the Extract from the Liposomes Based on TPC and TFC

The release from the liposomes loaded with *Glycyrriza glabra* L. was carried out by using a dialysis method. The procedure was as follows: an aliquot (800 µL) from the dispersion was added into a dialysis tube (D-Tube ^TM^ Dialyser Midi, MWCO 3.5 kDa, Sigma Aldrich) and incubated with 20 mL physiological solution (B. Braun Melsungen AG, Melsungen, Germany) and incubated at 37 °C and 110 rpm in a shaker–incubator. Volumes of 2 mL were drawn from the medium at pre-set time points as it was immediately replenished with fresh physiological solution. The TFC or TPC values in the samples were estimated by UV–vis spectroscopy based on the relevant calibration curves (according to the procedure in Section 2.2.2). Release of the extract from the liposomes based on TPC and TFC was expressed in quercetin or gallic acid equivalents.

#### 2.2.5. Safety Test

Cytotoxicity/phototoxicity was assessed by the BALB/3T3 neutral red uptake assay [56,57]. In this assay, 96-well microtiter plates were seeded at a density of 1 × 10^4^ cells/100 µL/well and were incubated for 24 h. Test compounds were administered at concentrations varying from 0.04 to 10 mg/mL. In the phototoxicity tests, 96-well plates were first irradiated with a dose of 2.6 J/cm^2^ and then were incubated for an additional 24 h. After neutral red staining, washing, and desorption with H_2_O/ethanol/acetic acid = 50/49/1 solution, the absorption was measured on a TECAN microplate reader (TECAN, Grödig, Austria) at wavelength 540 nm. Cytotoxicity/phototoxicity was expressed as CC_50_ values (concentrations required for 50% cytotoxicity/phototoxicity) and was calculated using non-linear regression analysis (GraphPad Software 8, San Diego, CA, USA).

The cytotoxicity test was also applied to confluent HCT-8 cells that were treated with 0.1 mL/well of the test substances at decreasing concentrations. The cells were incubated at 37 °C and 5% CO_2_ for 5 days. After microscopic evaluation, the medium with the test compound was removed, and the cells were washed and incubated with neutral red, at 37 °C for 3 h. The cells then were washed with PBS and overlaid with 0.15 mL/well desorbing solution. OD (optical density) was recorded at 540 nm in a microplate reader (Biotek Organon, West Chester, PA, USA). The 50% cytotoxic concentration (CC_50_) was calculated as the dose that affects cell viability by 50% compared to untreated controls. Each sample was tested in triplicate with four wells per test sample. The maximum tolerable concentration (MTC) was also estimated as the compound concentration which does not affect the cells significantly, and they look like the cells in the control (no compounds added).

#### 2.2.6. Determination of Infectious Viral Titers

HCT-8 cells were cultivated in 96-well plates and after the formation of a confluent monolayer were infected with 0.1 mL viral suspension in tenfold falling dilutions. Following 1 h of adsorption, the non-adsorbed virus was removed and 0.1 mL/well-supporting medium was added to the cells. The plates were incubated at 33 °C and 5% CO_2_ in the HERA cell 150 CO_2_ incubator (Radobio Scientific Co., Ltd., Shanghai, China) for 5 days. The infectious viral titer was determined by microscopic monitoring of the cellular monolayer and cytopathic effect (CPE) estimation. Visually defined CPE was confirmed by coloring with the dye neutral red (NRU). The development of the CPE, as a percentage of controls, was calculated for each dilution of the virus [56].

#### 2.2.7. Antiviral Activity Assay

The cytopathic effect (CPE) inhibition test was used for the assessment of the antiviral activity of the tested samples. Confluent cell monolayer in 96-well plates was infected with 100 cell culture infectious dose 50% (CCID_50_) in 0.1 mL (coronavirus OC43 strain). After 120 min of virus adsorption, the tested sample was added in various concentrations and the cells were incubated for 5 days at 33 °C and 5% CO_2_. The cytopathic effect was determined using an NRU assay and the percentage of CPE inhibition for each concentration of the sample was calculated following an established protocol and as described previously [18,19].

#### 2.2.8. Virucidal Assay

HCoV-OC43 (CCID_50_) viral suspension was mixed with the tested extracts at their maximal tolerable concentration (MTC) at a 1:1 ratio in a final volume of 1 mL. The contact samples were then incubated at 20 °C for 15, 30, 60, 90, and 120 min. The residual infectious virus content in each sample for each time point was determined by the endpoint dilution method of Reed and Muench [56] and the reductions in viral titer in Δlgs were calculated as compared to the untreated controls.

#### 2.2.9. Effect on the Viral Adsorption

Twenty-four-well plates containing HCT-8 cell monolayer were pre-cooled to 4 °C and inoculated with CCID_50_ of HCoV-OC43. In parallel, they were treated with the tested samples at their MTC and incubated at 4 °C for the time of virus adsorption. At various time intervals in the different samples (15, 30, 60, 90, and 120 min), the cells were washed with PBS to remove both the compound and the unattached virus, and then the cells were covered with a support medium and incubated at 33 °C and 5% CO_2_ for 48 h. After freezing and thawing three times, the infectious viral titer of each sample was determined by the final dilution method. Δlgs were estimated compared to the viral control (untreated with the compounds). Each sample was prepared in four replicates.

#### 2.2.10. Statistical Analysis

Non-linear regression analysis was performed for CC_50_ and IC_50_ calculation (GraphPad Software, San Diego, CA, USA). The values were presented as means ± SD from three independent experiments.

## 3. Results

### 3.1. Characterization of the Physicochemical Properties of the Produced Liposomes

Table 2 represents the results for the size and ζ-potential of the unloaded and extract-loaded liposomes. The determined electrokinetic potential for the unloaded control is higher compared to the structures carrying extracts.

It is well-known that liposomes are thermodynamically unstable and in order to improve their stability, chitosan is superficially layered on the produced structures. The overcompensation of surface charge results from the electrostatic interaction between the liposomes and the oppositely charged polymer. The positive value of the ζ-potential of the structures slightly increases with the elevation of chitosan concentration. Figure 1 represents the variation of the ζ-potential of the liposomes loaded with *Glycyrrhiza glabra* L. in the presence of different concentrations of chitosan. The electrokinetic behavior of the vesicles loaded with the other plant extracts is similar (the results are presented in the Appendix A).

The addition of the polymer solution (dissolved in diluted HCl in order to ensure maximum charge of the polysaccharide) to the liposomal dispersion produces a significant increase in the electrical conductivity of the mixture, especially at a higher chitosan concentration (ionic strength is about 0.02 M). Therefore, liposomes stabilized with chitosans at a concentration of 0.1 mg/mL were selected for further antiviral analyses. The results show that at this concentration of polymer, the dispersion is stable and the conductivity values are close to the ones for the initial dispersion of liposomes (ionic strength is about 10^−4^ M).

Table 3 represents the registered diameter of loaded structures in stabilized dispersion (at 0.1 mg/mL chitosan). The data indicates that the size distribution of the chitosan-covered liposomes is not monodisperse and in some of the samples, the standard deviation is high. Moreover, no correlation between liposomal size and the characteristics of chitosan (such as molecular weight) was observed. Further, an increase in the chitosan concentration results in a slight increase in the liposomes’ size.

### 3.2. Determination of the Amount of Encapsulation Extracts Based on TFC and TPC

Plant extracts are multi-component systems and their content is not constant and depends on many environmental factors. The total polyphenols (TPC) and total flavonoids (TFC) contents of the same extracts used in the present study were reported in our previous work but without samples’ filtration [18]. The total flavonoid content (TFC) and polyphenol content (TPC) are expressed in quercetin and gallic acid equivalents, respectively, using appropriate colorimetric methods and spectrophotometric analysis.

The experimental results indicate that the encapsulation efficacy was ca. 100% for all of the investigated samples. It was supposed that there was a significant affinity of the polyphenolic substances to the lipid bilayer of the liposomes. However, a significantly lower encapsulation efficiency was registered for the liposomes loaded with *Sambucus nigra* (Table 4).

Typically, the encapsulation efficiency of active molecules in liposomes is high which results from their unique structure. That is why the registered extremely high encapsulation efficiencies of quercetin and gallic acid are not surprising.

### 3.3. Release of TFC and TPC from the Liposomes Based on TPC and TFC

The release of compounds from *Glycyrriza glabra* L.-loaded liposomes in a physiological solution at 37 °C was measured by a dialysis method. The concentrations of free quercetin and gallic acid in the aliquots were estimated by UV–vis spectroscopy by using appropriate calibration curves (according to the same procedure as for the estimation of encapsulation efficiency). The liposomes loaded with this extract were selected based on previous microbiological experiments. Spectroscopy revealed that, after 24 h in physiological solution, the free concentration of compounds corresponds to about 42% (3.45 µg/mL) (quercetin) and 15% (0.51 mg/mL) (gallic acid). The implementation of experiments with a longer duration in physiological solution at 37 °C was hindered because of the occurrence of turbid sludge and spoilage of the sample.

### 3.4. Safety Testing

The compounds were studied for safety by an in vitro 3T3 NRU test. The cells were incubated with loaded liposomes at an extract concentration from 0.04 to 10 mg/mL for 24 h at 37 °C, 5% CO_2_, and 95% humidity. It is important to note that these dilutions of the stock dispersion are obtained in accordance with the amount of extracts used for the formation of liposomes. The cytotoxicity/phototoxicity expressed in % relative to the negative control was determined. Dose–response dependence was observed for all extracts. In the phototoxicity experiments, we used the phototoxic compound chlorpromazine as a positive control. The obtained results are shown in Figure 2. The results for liposomes loaded with extracts are compared with a control sample of liposomes loaded with pure substance REM which has well-known antiviral and cytotoxicity properties. Data are presented in Figure 2 and Table 5, Table 6 and Table 7 and serve only for comparison.

CC_50_ values (50% cytotoxic/phototoxic concentration) were obtained by non-linear regression analysis (Table 5) and were used to calculate the photo irritation factor (PIF) for each test compound, according to the formula: PIF = CC_50_ − Irr/CC_50_ + Irr. The PIF value shows us the probability test substance to cause a phototoxic effect (PIF < 2 not phototoxic, PIF ≥ 2 < 5 probable phototoxicity, and PIF ≥ 5 phototoxic). For unloaded liposomes, the calculated PIF was 5.75, which shows phototoxicity properties. Low phototoxicity was observed at L_REM_. No phototoxic effects were observed with liposomes containing plant extracts.

Before the determination of antiviral activity of all the tested samples, their cytotoxicity against the cell line, permissive to viral replication, was estimated. The information obtained from cytotoxicity assays allowed the antiviral experiments to be carried out at non-toxic concentrations of the products and the effect reported was not the result of toxicity.

Following the assessment of individual cytotoxicity and antiviral activity of medicinal plant extracts in our previous research [19], this study aimed to investigate their cytotoxic and antiviral capacities after they have been inserted into liposomes. The influence of the liposomes containing medicinal plant extracts on the viability of HCT-8 cells was determined. Comparing the cytotoxicity properties from our present and previous results, lower cytotoxicity was observed for most of the extract liposomes compared to the non-loaded extract values. Of the five examined liposomes with extracts, the lowest cytotoxicity was shown by L_SN_ with CC_50_ = 2350.0 µg/mL, and the highest by L_PR_ with CC_50_ = 1528.6 µg/mL (Table 6).

It is obvious that the extracts included in the composition of liposomes increased their effectiveness several times against coronavirus infection. The effect of the *A. hippocastanum* extract was increased nearly 16-fold, of the *P. reptans* extract more than 15-fold, that of the *G. glabra* root was 8-fold higher, the activity of the *A. sativum* extract had a 6-fold increase, and the *S. nigra* extract was 5-fold more effective than the native preparation. When remdesivir was introduced into the liposomes, its activity increased nearly 3-fold (Table 7).

After establishing the inhibitory potentials of the liposomes containing natural extracts against the HCoV-OC43 coronavirus strain, we set out to verify whether the effect was observed only in the infected replication-competent host cells or if the liposomes also affected the extracellular virions. The virucidal influence of empty and extract-containing liposomes was determined at time intervals varying from 15 min to 120 min. The obtained results indicate that none of the liposome samples discredited the HCoV-OC43 virions’ infectivity (Table 8).

In order to provide an additional verification that liposomes carrying natural extracts exert their antiviral effect only intracellularly, we tested whether the liposomes act on the initial step of the viral cycle, i.e., on the adsorption to the host cells. From the results presented in Table 9, it can be concluded that this stage of viral reproduction is also not affected by the presence of liposomes, either empty or containing extracts, which once again proves that the anti-coronavirus effect occurs only inside the cell.

## 4. Discussion

### 4.1. Unloaded and Extract-Loaded Liposomes

According to the experimental results presented in Table 2, the determined electrokinetic potential of unloaded liposomes is higher compared to the extract-loaded structures. This might be partially explained by the incorporation of substances from the extracts in the head group part of the lipid bilayer due to the electrostatic interactions. Since the lipid used (DOPC) is known to be a zwitterion, both positive and negative components contained in plant extracts could interconnect to the liposome. As a result, the overall electrokinetic potential of the liposomes is expected to be decreased. This suggestion is also in line with the results for the size of the liposomes. The incorporation of the extracts causes a decrease in the liposomal size compared to the unloaded liposomes.

As previously mentioned, polyphenols are involved predominantly in the lipid bilayer of the liposomes rather than in their inner core. Hence, the registered size of loaded liposomes is higher compared to the unloaded ones [39,48,50]. However, membranes formed from DOPC and loaded with quercetin have been reported to have a slight decrease in the bilayer thickness compared to the unloaded liposomes [58]. In this study, we detected a slight decrease in the hydrodynamic diameter of the extract-loaded liposomes compared to the unloaded ones. It should also be considered that the introduction of extract components into the hydrophobic part of the bilayer may reduce the fluidity common for pure DOPC (because of its unsaturated chains), and thus produce a more tightly packed bilayer.

A significant decrease in size is observed for liposomes carrying *Sambucus nigra* and *Allium sativum* extracts which probably correlates with the lower encapsulation amount of quercetin (Section 3.2). From the results presented in Table 2 and Table 4, the L_AS_ liposomes loaded with *Allium sativum* (*garlic*) have different characteristics. The L_AS_ liposomes possess significantly smaller dimensions compared to the other types of extract-loaded liposomes and the registered electrokinetic potentials are similar to the ones of the extract-free particles. Moreover, the encapsulation efficiency estimated against quercetin and gallic acid is lower compared to the other types of loaded liposomes. The experimental data also indicate that their cytotoxicity and phototoxicity is significantly lower compared to the other type of liposomes (Figure 2).

Particularly interesting is the subject of lipid–polyphenol interactions, which is related to the polyphenol position with respect to the liposomal membrane. Although it depends on the concrete structure of the components, there are some general factors that deserve mentioning. Firstly, the forces that define the intermolecular behavior include for the most part H-bonding between the polyphenol OH groups and the lipid headgroups (–PO_3_, C=O), which hang strongly on the OH number and the degree of polyphenol penetration into the bilayer, either incorporated into the hydrophilic lipid portion (favorable for H-linking), or deeper in the acyl chains (inhibits H-bonding for spatial reasons); the latter is attributed to hydrophobic interactions between the corresponding aqua-repelling parts of both components; van der Waals forces, when dipole–dipole or charge–dipole interactions play an important role since a number of flavonoids are characterized with significant dipole moments [59]. In general, H-linking and in particular the bigger number of polyphenol OH^−^ groups, as well as the van der Waals forces, support polyphenol anchoring at the hydrophilic part of the bilayer, while polyphenol hydrophobicity is a precondition for sinking into the hydrocarbon portion of the lipid membrane [48,60]. Certain configurational features of polyphenols influence their behavior with respect to lipids: when the structure is more flat-like shaped, for instance, by two coplanar aromatic rings (common for quercetin and quercetin-like flavonoids) or double bonds, it would cause deeper incorporation into the lipid layer, while the presence of glucoside groups, on the contrary, would inhibit it [58]. Factors of electrostatic origin also can control the mutual position: for example, the electric charge of the lipids favors the polyphenol location at the level of the headgroups and enhanced environmental salt concentration promotes its greater infiltration into the liposome bilayer because of the shielding effect of the medium electrolyte over the lipid charge [61]. Once and again, the polyphenol-induced rearrangement in the liposome membrane is concentration-dependent and reversible, which facilitates a successful release of the active component at a later stage of the extract delivery [62].

The use of different types of drug delivery systems aims to achieve controlled and sustainable release of the active components, as well as to manage targeted delivery to specific cells and tissues and thus to reduce the side effects of drugs by widening the therapeutic range between the lowest effective and toxic concentrations [63]. Chitosan is a naturally occurring biopolymer with high biocompatibility and absent toxicity, as both low- and high-molecular-weight chitosan is easily metabolized in the organism and easily removed by renal clearance [64]. Chitosan is preferred in the formulation of nanocarriers due to its cationic nature, allowing interactions with anionic drugs or compounds, as well as its mucoadhesive properties, which ensures greater tissue penetration and delivery of a greater quantity of the therapeutic it carries [61,64]. Microparticles, as well as nanoparticles (NPs), composed of chitosan cross-linked with tripolyphosphate, were prepared for the delivery of acyclovir. The system demonstrated biocompatibility, bioadhesive properties, and potential as a skin permeation enhancer. Furthermore, chitosan-based particles caused less tissue damage and only moderate irritation as assessed by the snail mucosal irritation (SMI) test [62,65]. Another system for the delivery of acyclovir through the skin was created, based on chitosan–tripolyphosphate NPs with good chemical stability. If the chitosan content of the NP is higher, a greater amount of acyclovir penetrates through the porcine skin. Cutaneous diffusion is improved with acyclovir particles, especially those with high chitosan content [66].

### 4.2. Antiviral Activity of Extract-Loaded Liposomes

The results of an in vitro study showed that the chitosan/siRNA nanoparticles were efficiently taken up by Vero cells, resulting in the inhibition of influenza virus replication. In addition, nasal delivery of siRNA via a chitosan nanoparticle complex has antiviral effects and significantly reduces mortality in BALB/c mice [67]. Chitosan nanoparticles for the delivery of foscarnet maintained the antiviral activity of the released drug when tested in vitro against lung fibroblast (HELF) cells infected with HCMV strain AD-169. Moreover, the nanoparticles showed no toxicity on uninfected HELF cells [68]. Statistically engineered mucoadhesive chitosan–alginate nanoparticles (MCS-ALG-NPs) as a novel carrier for favipiravir (FVR) to the porcine epidemic diarrhea virus (PEDV) model, which serves as a SARS-CoV-2 surrogate, demonstrated superior adhesion, improved penetration, accumulation in nasal mucosa, and an over 35-fold increased inhibition of viral replication compared to free FVR [69].

Traditional medicine has been using natural products for the treatment of various infectious diseases for centuries. However, due to the difficult penetration of a large part of the biologically active molecules through the cell membrane into the cell, their effect is limited. The introduction of natural substances into nanocarriers significantly increases their bioavailability and effectiveness. Curcumin-encapsulated chitosan nanoparticles exhibited lower toxicity in Crandell–Rees feline kidney (CrFK) cells and enhanced antiviral activity with a selective index (SI) value three times higher than that of curcumin against feline infectious peritonitis virus (FIPV) [70].

Encapsulation of *Nigella sativa* extract in nanoparticles formed between chitosan and a water-soluble fraction of Persian gum enhanced the antiviral activity of the extract against infectious bronchitis virus (IBV), known as avian coronavirus [71]. When examining chitosan nanoparticles prepared using a chemical cross-linking agent and containing bee venom (BV), the results showed that crude BV had a mild anti-MERS-CoV with a selective index (SI = 4.6), followed by that of chitosan NPs, which showed moderate anti-MERS-CoV with SI = 8.6. Meanwhile, the nanocomposite showed promising anti-MERS-CoV properties with SI = 12.1 [72]. Silymarin–chitosan formulations were also investigated for their antiviral potential against SARS-CoV-2 and ADV-5, using in silico and in vitro approaches, which showed improvements in antiviral activity, bioavailability, and physicochemical properties. The increased antiviral activity may occur through blockage of the host receptor ACE2, thereby preventing the virus from attachment and entry into the cell [73].

However, our results contradict this statement as liposomes containing extracts did not affect the step of virus adsorption to the host cell. The action of the medicinal-plant-extract-loaded nanostructures takes place only after the vesicles deliver the substances carried inside the cells, where they are released, exerting their biological effect only intracellularly. The data obtained suggest that the utilization of chitosan-coated nanocarriers, transporting substances with anti-coronavirus action, increases the activity of the particular viral inhibitor in general, which is in accordance with other studies on different but still close enough in structure to the liposomes used herein.

## 5. Conclusions

This study reports the properties of liposomal particles loaded with natural extracts from medical plants (*Glycyrriza glabra* L.; *Sambucus nigra*; *Aesculas hippocastanum*; *Potentilla reptans*; *Allium sativum*) used in traditional medicine for centuries. Aqueous solutions of the extracts were entrapped into the liposomes and the stability of the produced delivery systems was improved predominantly by the electrostatic adsorption of chitosan. The results presented prove that the use of extract-loaded liposomes increases several times the anti-coronavirus activity of a given natural extract. The effect observed with the extract of *A. hippocastanum* was the most distinct and it was obvious that it was exerted only during the intracellular stages of viral replication but not on the extracellular virions or on their adsorption to the host cell, which proves that the extract-loaded liposomes act only as carriers of an active substance that acts mainly intracellularly after its delivery to the target cell. Our future studies will aim to investigate the antiviral activity of metabolites contained at the highest concentration in the extracts, as well as to determine at which particular step of viral replication the observed effect takes place.

## Figures and Tables

**Figure 1 life-14-01180-f001:**
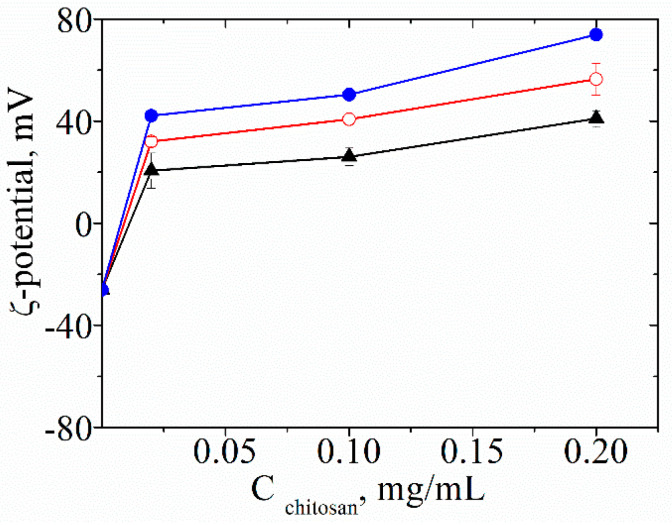
The ζ-potential of liposomes (L_GGL_) loaded with a plant extract from *Glycyrriza glabra* L. as a function of the concentration C_chitosan_ of three different chitosans (described in Section 2.1.3) added to the dispersion: (●) CS-L, (○) CS-H, and (▲) COS.

**Figure 2 life-14-01180-f002:**
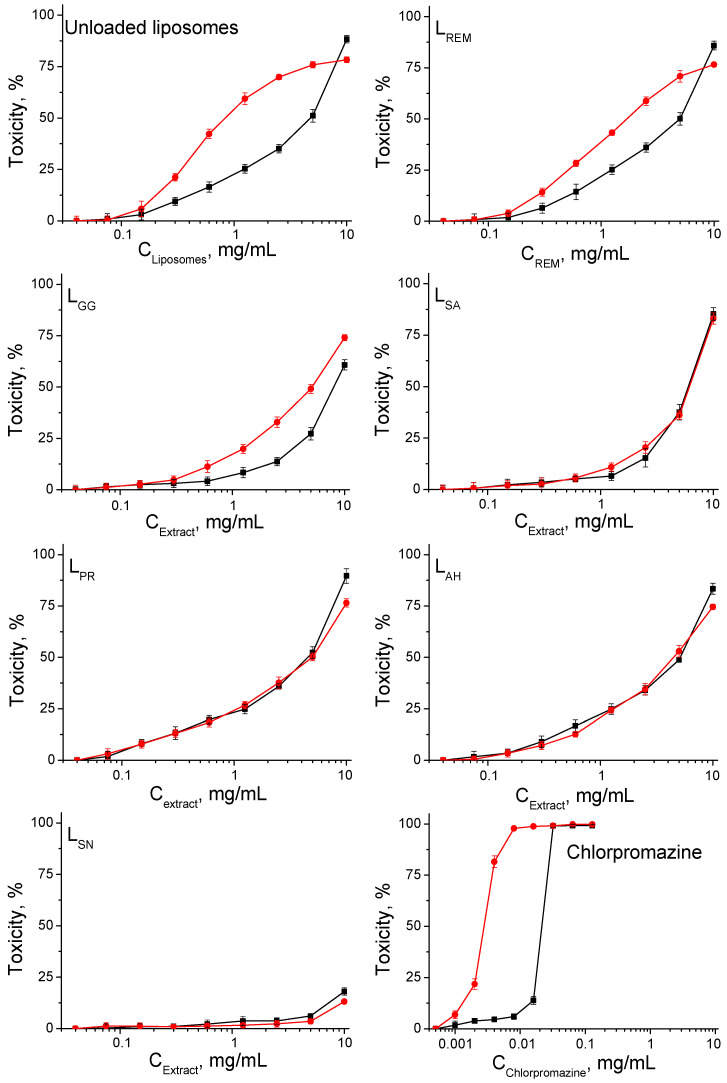
Cyto-(■) and phototoxicity (●) dose–response curves determined in BALB/3T3 cells. Data are means ± standard deviation from three independent experiments, *n* = 6. The liposomes are stabilized with CS-L (0.1 mg/mL).

**Table 2 life-14-01180-t002:** Hydrodynamic diameter D, polydispersity index PDI, and ζ-potential) of the liposomes loaded with a plant extract: L_SN_ (*Sambucus nigra*), L_AS_ (*Allium sativum*), L_PR_ (*Potentilla reptans*), L_AH_ (*Aesculas hippocastanum*), L_GGL_ (*Glycyrriza glabra* L.).

Liposomes	D *, nm	PDI	ζ-Potential, mV
L_SN_	197.6 ± 85.1	0.746	−44.2 ± 1.0
L_AS_	50.8 ± 22.9	0.713	−49.1 ± 1.5
L_PR_	229.5 ± 14.6	0.708	−36.5 ± 1.7
L_AH_	221.2 ± 15.2	0.401	−27.9 ± 2.4
L_GGL_.	261.2 ± 9.8	0.211	−26.1 ± 1.8
L_unloaded_	267.6 ± 16.2	0.153	−50.2 ± 2.1

* sizes are determined by a mean intensity.

**Table 3 life-14-01180-t003:** Hydrodynamic diameter, D, and polydispersity index, PDI, of chitosan-stabilized liposomes loaded with plant extracts: L_SN_ (*Sambucus nigra*), L_AS_ (*Allium sativum*), L_PR_ (*Potentilla reptans*), L_AH_ (*Aesculas hippocastanum*), L_GGL_ (*Glycyrriza glabra* L.). Chitosan concentration is 0.1 mg/mL.

Liposomes.	COS		CS-L		CS-H	
	D *, nm	PDI	D *, nm	PDI	D *, nm	PDI
L_SN_	361.3 ± 110.5	0.48	568.3 ± 150.7	0.39	146.1 ± 78.7	0.53
L_AS_	229.8 ± 80.5	0.44	103.1 ± 32.4	0.37	488.7 ± 108.	0.36
L_PR_	361.3 ± 86.5	0.43	229.8 ± 57.5	0.36	169.9 ± 50.1	0.47
L_AH_	187.5 ± 4.5	0.30	187.6 ± 2.3	0.32	222.8 ± 5.1	0.31
L_GGL_.	191.1 ± 6.6	0.23	187.5 ± 4.5	0.22	232.5 ± 22.2	0.35
L_unloaded_						

* sizes are determined by a mean intensity.

**Table 4 life-14-01180-t004:** Determination of the encapsulation efficiency of the extracts based on TFC and TPC (expressed in quercetin and gallic acid equivalents), EE%, in the liposomes loaded with a plant extract: L_SN_ (*Sambucus nigra*), L_AS_ (*Allium sativum*), L_PR_ (*Potentilla reptans*), L_AH_ (*Aesculas hippocastanum*), L_GGL_ (*Glycyrriza glabra* L.). The concentrations $C_{extract}^{quercetin\;equivalents}$ and $C_{extract}^{gallic\;acid\;equivalents}$ correspond to the total amounts of the compounds in the extract solutions.

Plant Extract	$C_{extract}^{quercetin\;equivalents}$ µg/mL	EE _(TFC)_ %	$C_{extract}^{gallic\;acid\;equivalents}$ mg/mL	EE _(TPC)_ %
L_SN_	0.1700	99	1.53	≈100
L_AS_	0.0014	70	0.57	98
L_PR_	10.7959	≈100	2.54	≈100
L_AH_	14.7531	≈100	2.10	≈99.5
L_GGL_.	8.2132	≈100	3.40	≈100

**Table 5 life-14-01180-t005:** CC_50_ values of mean and photo irritation factors.

Sample	Mean CC_50_ ± SD (mg/mL)		PIF *
	−Irr	+Irr	
Unloaded liposomes	2.364 ± 0.2434	0.411 ± 4.03	5.75
L_REM_ **	2.419 ± 0.2306	0.831 ± 2.57	2.91
L_GG_	4.002 ± 0.2005	2.543 ± 16.76	1.57
L_AS_	2.993 ± 0.1001	3.071 ± 5.1	0.97
L_PT_	2.269 ± 0.2151	2.459 ± 13.8	0.92
L_AH_	2.562 ± 0.0513	2.245 ± 23.73	1.14
L_SN_	>10	>10	-
Chlorpromazine ***	0.022 ± 0.003	0.003 ± 0.0006	7.33

* PIF (photo irritation factor), PIF < 2 not phototoxic, PIF ≥ 2 < 5 probable phototoxicity, PIF ≥ 5 phototoxic; ** control sample; *** positive control.

**Table 6 life-14-01180-t006:** HCT-8 cytotoxicity of liposomes containing natural extracts stabilized with CS-L (0.1 mg/mL) against the HCT-8 cell line.

Sample	HCT-8 Cell Line	
	CC_50_ * Mean ± SD ** [µg/mL]	MTC *** [µg/mL]
Unloaded liposomes	≥1000	≥1000
REM	2500.00 ± 4.3 ^#^	1000.0 ^#^
L_REM_	2358.0 ± 25.2	1195.0
*Extract* (*A. hippocastanum*)	1420.0 ± 46.2 ^##^	800.0 ^##^
L_AH_	1839.6 ± 28.7	900.0
*Extract* (*A. sativum*)	1880.0 ± 55.7 ^##^	1200.0 ^##^
L_AS_	2055.3 ± 37.2	1300.0
*Extract* (*S. nigra*)	1900.0 ± 48.3 ^##^	1000.0 ^##^
L_SN_	2350.0 ± 38.7	1150.0
*Extract* (*G. glabra*)	1820.0 ± 24.5 ^##^	1000.0 ^##^
L_GG_	1817.0 ± 27.3	1100.0
*Extract* (*P. reptans*)	1880.0 ± 37.1 ^##^	200.0 ^##^
L_RP_	1528.6 ± 26.8	350.0

* CC_50_—cytotoxic concentrations 50%; ** SD—standard deviation; *** MTC—maximum tolerable concentration; ^#^ the result was presented in a previous study [18]; ^##^ the result was presented in a previous study [19].

**Table 7 life-14-01180-t007:** Antiviral activity of liposomes containing natural extracts stabilized with CS-L (0.1 mg/mL) against the replicative cycle of the human coronavirus strain HCoV-OC43.

Sample	HCoV-OC43	
	IC_50_ * Mean ± SD ** (µg/mL)	SI ***
Unloaded liposomes	-	-
REM	12.5 ± 0.9 ^#^	200.0 ^#^
L_REM_	4.3 ± 0.8	548.3
*Extract* (*A. hippocastanum*)	380.0 ± 9.5 ^##^	3.7 ^##^
L_AH_	31.2 ± 2.4	58.96
*Extract* (*A. sativum*)	900.0 ± 18.5 ^##^	2.1 ^##^
L_AS_	151.5 ± 8.2	13.56
*Extract* (*S. nigra*)	950.0 ± 32.7 ^##^	2.0 ^##^
L_SN_	215.0 ± 7.3	10.93
*Extract* (*G. glabra*)	400.0 ± 12.5 ^##^	4.5 ^##^
L_GG_	46.5 ± 3.9	39.1
*Extract* (*P. reptans*)	890.0 ± 17.3 ^##^	2.1 ^##^
L_PR_	47.5 ± 3.3	32.2

* IC_50_—inhibitory concentration 50%; ** SD—standard deviation; *** SI—selectivity index is calculated from the CC_50_/IC_50_ ratio.; ^#^ the result was presented in a previous study of ours [18]; ^##^ the result was presented in a previous study of ours [19].

**Table 8 life-14-01180-t008:** Virucidal activity of liposomes containing extracts of medicinal plants stabilized with CS-L (0.1 mg/mL) against coronavirus virions from strain HCoV-OC43.

Sample	Δlg				
	15 min	30 min	60 min	90 min	120 min
Unloaded liposomes	0.25	0.25	0.25	0.25	0.25
L_AH_	0.25	0.25	0.50	0.50	0.50
L_AS_	0.25	0.25	0.25	0.25	0.25
L_SN_	0.15	0.15	0.25	0.25	0.25
L_GG_	0.25	0.25	0.25	0.25	0.25
L_PR_	0.15	0.15	0.15	0.25	0.25
70% ethanol	5.00	5.00	5.00	5.00	4.75

**Table 9 life-14-01180-t009:** Influence of liposomes containing extracts of medicinal plants and stabilized with CS-L (0.1 mg/mL) on the adsorption of human coronavirus strain HCoV-OC43 to HCT-8 cells.

Sample	Δlg				
	15 min	30 min	60 min	90 min	120 min
Unloaded liposomes	0.00	0.15	0.25	0.25	0.25
L_AH_	0.00	0.00	0.25	0.25	0.25
L_AS_	0.00	0.25	0.25	0.25	0.25
L_SN_	0.00	0.15	0.15	0.15	0.15
L_GG_	0.15	0.15	0.15	0.25	0.25
L_PR_	0.00	0.15	0.15	0.15	0.15

## Data Availability

The raw data supporting the conclusions of this article will be made available by the authors on request.

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
