# Peer review of "Anti-Coronavirus Activity of Chitosan-Stabilized Liposomal Nanocarriers Loaded with Natural Extracts from Bulgarian Flora"

_life, 2024, doi:10.3390/life14091180_

Round 1

Reviewer 1 Report

Comments and Suggestions for Authors

 The authors reported a very interesting study. However, the reviewer has the following comments and concerns:

  1. The abstract is missing from the manuscript.
  2. Professional English-language editing is necessary. For example, “The our work” (line 38) needs correction.
  3. Avoid single-sentence paragraphs.
  4. In section 2.1 (Materials), certain sections can be merged into one.
  5. In section 2.2 (Methods), line 150: How was the liposome concentration estimated?
  6. Figure 1 doesn’t provide much useful information. It could be revised to show the extract-loaded liposome preparation process.
  7. In Equation (1), change “.” to “×” and “100” to “100%.” Revise Equation (2) accordingly.
  8. In section 2.2.6, is there a reference for the procedure? It seems unlikely that cells would be incubated with test substances for 5 days.
  9. Define all abbreviations, such as PDI.
  10. In Table 3, data is missing for the row labeled "Lunloaded."
  11. The determination of EE% is confusing. Were quercetin and gallic acid encapsulated directly, or were they used to determine the amount of extracts?
  12. Both the safety test and cytotoxicity assay are toxicity tests.
  13. In section 3.4, what was the difference between the cytotoxicity and phototoxicity positive controls? Was it only with or without radiation?
  14. In Tables 8 and 9, most data are listed as 0.15 and 0.25. Please double-check.
Comments on the Quality of English Language

Professional English-language editing is necessary.

Author Response

  1. Remark: The abstract is missing from the manuscript.

Response:  In the currently uploaded version, the abstract is included in the manuscript. I assume that its absence in the previous version is some kind of technical error.

  1. Remark: Professional English-language editing is necessary. For example, “The our work” (line 38) needs correction.

Response:   The manuscript was completely edited by an English-speaking philologist.

  1. Remark: Avoid single-sentence paragraphs.

Response:    We thank the esteemed reviewer for the recommendation. Wherever possible in the manuscript single-sentence paragraphs have been avoided.

  1. Remark: In section 2.1 (Materials), certain sections can be merged into one.

Response:   Sections 2.1.1. and 2.1.2 are merged into one, as well as sections 2.2.5. and 2.2.6. From the methodology section, we also combined 2.2.5. and 2.2.6. in one section.

  1. Remark: In section 2.2 (Methods), line 150: How was the liposome concentration estimated?

Response:   The concentration of the liposomes in dispersion was calculated from the values of the lipid concentration, size of the liposomes and the head size of the lipid molecule in solution at full packaging of molecules (spheres) on the flat surface (0.636×10-18 m2 [Berkowitz ML, Vácha R (2012), Acc Chem Res 45:74–82]. The determined concentration of liposomes corresponds to a structures with lipid bilayer.

  1. Remark: Figure 1 doesn’t provide much useful information. It could be revised to show the extract-loaded liposome preparation process.

Response:   Figure 1 is removed and the numbers of figures are changed.

  1. Remark: In Equation (1), change “.” to “×” and “100” to “100%.” Revise Equation (2) accordingly.

Response:  Equation 1 is revised and Equation 2 is removed.

  1. Remark: In section 2.2.6, is there a reference for the procedure? It seems unlikely that cells would be incubated with test substances for 5 days.

Response:   The different strains of the Coronavirus are more difficult to cultivate in vitro than viruses from other families. The cytopathic effect is produced very slowly and it is necessary to wait a little longer than with other viruses. According to data from research and other authors, in order for a clear cytopathic effect to appear in the case of coronavirus, it is necessary to wait between 3 to 10 days. We have chosen to measure the effect on the 5th day, when the effect is clearly visible. Accordingly, this is also the period for treatment with the relevant extracts, because there is no way to report antiviral activity of any product before the virus itself has manifested its cytopathic effect. I present several references in which other authors indicate the period from 3 to 10 days to consider the cytopathic effect.

O'Keefe BRGiomarelli B, Barnard DL, Shenoy SR, Chan PKS, McMahon JB, Palmer KEBarnett BW, Meyerholz DK, Wohlford-Lenane CL, McCray PB 2010. Broad-Spectrum In Vitro Activity and In Vivo Efficacy of the Antiviral Protein Griffithsin against Emerging Viruses of the Family Coronaviridae. J Virol 84:.

https://doi.org/10.1128/jvi.02322-09

Kaye M. SARS-associated coronavirus replication in cell lines. Emerg Infect Dis. 2006 Jan;12(1):128-33. doi: 10.3201/eid1201.050496. PMID: 16494729; PMCID: PMC3291385.

Hu, Y., Meng, X., Zhang, F., Xiang, Y., & Wang, J. (2021). The in vitro antiviral activity of lactoferrin against common human coronaviruses and SARS-CoV-2 is mediated by targeting the heparan sulfate co-receptor. Emerging Microbes & Infections, 10(1), 317–330. https://doi.org/10.1080/22221751.2021.1888660

  1. Remark: Define all abbreviations, such as PDI.

Response: The abbreviations in Table 2, Figure 2 and Table 3 are corrected.

  1. Remark: In Table 3, data is missing for the row labeled "Lunloaded."

Response:  Table 3 is revised.

  1. Remark: The determination of EE% is confusing. Were quercetin and gallic acid encapsulated directly, or were they used to determine the amount of extracts?

Response:   The estimated EE% corresponds to quercetin and gallic acid concentration in the liposomes.

  1. Remark: Both the safety test and cytotoxicity assay are toxicity tests.

Response:   In its essence, it is exactly so. Therefore, according to your 4th remark, we combined the two tests as "Safety test", because both tests determine the safe (non-toxic) concentrations of the test products, but in one case the non-toxic range of the test product is determined on the day of reporting the cytopathic effect (for to prove that the obtained effect is not the result of toxicity of the product under investigation). In the other case, different concentrations of the product are tested again, but in a more complex setup. Two sets of samples are prepared. Some samples are irradiated, and the others are incubated in the dark, and when comparing the toxicity between them, it can be seen whether the studied product exhibits greater cytotoxicity when irradiated. In this case, the reporting takes place on the 24th hour.

  1. Remark: In section 3.4, what was the difference between the cytotoxicity and phototoxicity positive controls? Was it only with or without radiation?

Response:   Three types of controls are used in determining phototoxicity: (i) non-irradiated and untreated with the test product (ie, untreated, non-irradiated healthy cells); (ii) a control containing irradiated cells but not treated (to trace only the effect of the light on the cells), and (iii) a control of cells not irradiated but treated with the product (that is, giving the true toxicity of the product in absence of light).

  1. Remark: In Tables 8 and 9, most data are listed as 0.15 and 0.25. Please double-check.

Response:   We are absolutely sure of the obtained results presented in Tables 8 and 9 and there is no error in their writing. These values ​​indicate the change in virus titer (Δlg) upon treatment with the respective product, compared to the titer of the untreated virus (virus control). When we have a decrease in the viral titer by Δlg < 1, we assume that the obtained difference is not statistically sufficient to say that there is an effect. That is, our results show that the examined liposomes containing extracts do not affect the extracellular virions and the stage of their adsorption to the cell. This is a positive result, because these liposomes are designed to carry the active component within them to the target (infected) cells, and not to have an effect outside the cell. This also proves the stability of the liposomes - because the extracts contained in them do not exert their effect on the virions (which in our previous study was shown to be significant), indicating that the extracts remain in the liposomes and no disintegration of the system composed of liposome-extract.

Reviewer 2 Report

Comments and Suggestions for Authors

The manuscript by Gyurova et al reports the preparation of 5 chitosan-stabilized liposomes containing natural extracts from 5 different medicinal plants and their activity against coronavirus. The hypothesis of the study was established and the observed data validates it. The methods and procedures were clearly presented. Table 1 is great and increases the importance of this manuscript. Many studies were reported to validate the formation of liposomes and their action mode. The study is solid and merits publication after the following minor corrections.

1. many minor grammatical errors need to be corrected. 

2. Fix the units for temperature, in many cases, there is no space between the value and the degree symbol and it needs a space. Also, see lines 238 and 243.

3. A picture of the liposomes will be great. 

4. Include in the conclusion which plant extract is better and your recommendation for future testing, 

5. the abstract is missing.

6. Clearly describe, how were the coronavirus and samples disposed of after the study?

7. Usemg/ mL on Figure 5

Comments on the Quality of English Language

Many minor grammatical errors required attention.

Author Response

  1. Remark: many minor grammatical errors need to be corrected. 

Response:   The text has been extensively revised and many grammatical errors have been corrected.

  1. Remark: Fix the units for temperature, in many cases, there is no space between the value and the degree symbol and it needs a space. Also, see lines 238 and 243.

Response:   We thank the esteemed reviewer for the remark. The entire manuscript has been reviewed and a space has been added to the temperature unit wherever it was omitted.

  1. Remark: A picture of the liposomes will be great. 

Response:   The quality of the TEM images is not good. That is why, there are no images in the manuscript.

  1. Remark: Include in the conclusion which plant extract is better and your recommendation for future testing, 

Response:   The additions suggested by the respected reviewer in the Conclusion have been made.

  1. Remark: the abstract is missing.

Response:   In the currently uploaded version, the abstract is included in the manuscript. I assume that its absence in the previous version is some kind of technical error.

  1. Remark: Clearly describe, how were the coronavirus and samples disposed of after the study?

Response:  All viral samples were first inactivated with 20% bleach. Thus, they stay for at least 24 hours in a specialized box for working with viruses, followed by additional inactivation by heat treatment and increasing the pressure in a special room of the Institute of Microbiology, BAS. An autoclave apparatus is used, which is only used for the disposal of contaminated material under conditions of 100 kPa pressure, at a temperature of 120 °C for 40 min.

  1. Remark: Usemg/ mL on Figure 5

Response:   The correction has been made.

Reviewer 3 Report

Comments and Suggestions for Authors

The authors aimed to investigate the antiviral activity of liposomes loaded with natural extracts from some common medicinal plants. The idea of the study is interesting, but the manuscript is needs a deep revision.

Major concerns:

1) In the manuscript, the authors stated that they quantified gallic acid and quercetin. This is incorrect. They used a non-specific spectrophotometric test. In fact, they measured the total flavonoid content (TFC), expressed in quercetin equivalent, and the antioxidant capacity, expressed as total phenolic content (TPC) in gallic acid equivalent. This should be corrected throughout the manuscript.

2) Lack of evaluation on whether the differences between the obtained values for tested samples were statistically significant (Tables).

3) It is not clear why a human colon carcinoma cell line assay was used for the antiviral test. The justification for selecting this type of cell line should be added.

4) In the Results section the Authors should describe only their own results. They  unnecessarily include elements of discussion and comparison with literature data in this section.

Other comments

1) In lines 38-42, the author states: “Our work will attempt to demonstrate the biological potentials of several flavonoid-rich extracts from medicinal plants…” The biological potential of the mentioned plants has been evidenced in many studies (some of which are cited here). This part is misleading and should be removed. The aim of the study is clearly presented in lines 77-83.

2) In the Introduction, the criteria for selecting the plants used in the study should be added. The expression “the polyphenol-containing Bulgarian medicinal plants” is not sufficient because polyphenols are present in all plants.

3) The placement of Table 1 in the Materials and Methods section is incorrect. Alternatively, it could be a part of the Introduction or Discussion to show the biological potential of the plants in the context of antiviral activity and justify the selection of the plants. Or it should be moved to Supplementary material

4) Line 143: correct: “rotationto form”

5) Figure 1 legend is misleading. See major comment regarding gallic acid and quercetin. Explanation for “DOPC” should be added.

6) 2.2.2. section: This section needs to be rewritten taking into account my comment regarding quantification of gallic acid and quercetin. Moreover, reference 57 is incorrect. In mentioned paper Authors did not use the spectrophotometric method.

7) Line 213: “the test compounds” should be replaced by “ the tested extracts”. The same for line 263, 388 – please check all text.

8) 3.2 section title. Title should be corrected to: “Determination of the Amount of Encapsulated Extracts”. This section needs to be rewritten taking into account my major suggestion (also Table 4, and 3.3. section).

9) “the total amount of TPC and TFC is ca. 5 %” – TPC and TFC is not express as a % but as an equivalent of gallic acid and quercetin, respectively. Lines 333-336 are unnecessary and should be removed.

10) Figures 3 and 4 are unnecessary and should be removed.

11) Table 6: Why was a specific value not provided for unloaded liposomes?

12) The Discussion section should be rewritten. Some parts  of the text describe the obtained results. The results related to cytotoxicity are not discussed. Section 4.2: Antiviral Activity: Lines 525-543 are not related to antiviral activity; they describe different types of drug delivery systems. This content should be moved to Section 4.1.

Author Response

Major concerns:

  • Remark: In the manuscript, the authors stated that they quantified gallic acid and quercetin. This is incorrect. They used a non-specific spectrophotometric test. In fact, they measured the total flavonoid content (TFC), expressed in quercetin equivalent, and the antioxidant capacity, expressed as total phenolic content (TPC) in gallic acid equivalent. This should be corrected throughout the manuscript.
  • Remark: Lack of evaluation on whether the differences between the obtained values for tested samples were statistically significant (Tables).

Response:  On questions 1) and 2) we completely agree with your remarks. Section 2.2.2., Section 2.2.4., Section 3.2. and Section 3.3. are rewritten.

  • Remark: It is not clear why a human colon carcinoma cell line assay was used for the antiviral test. The justification for selecting this type of cell line should be added.

Response:   Like any viral model, the coronavirus has certain cell lines on which it can develop a cytopathic effect. In our previous studies, we have also used other cell models such as MRC-5, but the cytopathic effect was weaker, so we switched to working with the HCT-8 cell line. It is one of the standard cell lines for in vitro studies conducted with coronavirus. I present several quotes of other teams who also used the same cell line.

Siragam V, Maltseva M, Castonguay N, Galipeau Y, Srinivasan MM, Soto JH, Dankar S, Langlois M. 2024. Seasonal human coronaviruses OC43, 229E, and NL63 induce cell surface modulation of entry receptors and display host cell-specific viral replication kinetics. Microbiol Spectr 12:e04220-23, https://doi.org/10.1128/spectrum.04220-23

Durbadal Ojha, Forrest Jessop, Catharine M. Bosio, Karin E. Peterson, Effective inhibition of HCoV-OC43 and SARS-CoV-2 by phytochemicals in vitro and in vivo, International Journal of Antimicrobial Agents, 2023, 62(3), 106893,

Yasuha Arai, Itaru Yamanaka,Toru Okamoto,Ayana Isobe, Naomi Nakai, Naoko Kamimura, Tatsuya Suzuki, Tomo Daidoji, Takao Ono, Takaaki Nakaya, Kazuhiko Matsumoto, Daisuke Okuzaki, Yohei Watanabe. Stimulation of interferon-β responses by aberrant SARS-CoV-2 small viral RNAs acting as retinoic acid-inducible gene-I agonists. iScience. 2023, 26(1), 105742

  • Remark: In the Results section the Authors should describe only their own results. They  unnecessarily include elements of discussion and comparison with literature data in this section.

Response:   We thank the esteemed reviewer for the remark. Unnecessary discussion items have been removed from the Results section.

Other comments

1 Remark:  In lines 38-42, the author states: “Our work will attempt to demonstrate the biological potentials of several flavonoid-rich extracts from medicinal plants…” The biological potential of the mentioned plants has been evidenced in many studies (some of which are cited here). This part is misleading and should be removed. The aim of the study is clearly presented in lines 77-83.

Response:   We thank the esteemed reviewer for the remark. Indeed, our statement is not correct. We have proven the antiviral activity of the extracts in our previous two (published) studies. We have corrected our statement.

2 Remark:  In the Introduction, the criteria for selecting the plants used in the study should be added. The expression “the polyphenol-containing Bulgarian medicinal plants” is not sufficient because polyphenols are present in all plants.

Response:   As mentioned in several places, the present development is a continuation of our previous studies. A group of medicinal plant extracts that are included in Mirta-Medicus Ltd products sold commercially as nutritional supplements designed to improve various health problems were investigated for antiviral activity. The extracts were not screened for antiviral activity until then, and this was the aim of our previous study. Those of them that showed anti-coronavirus activity were included in the present study to test whether their incorporation into nanoliposomes would increase their activity.

We have introduced the corresponding clarification in the manuscript.

3 Remark:  The placement of Table 1 in the Materials and Methods section is incorrect. Alternatively, it could be a part of the Introduction or Discussion to show the biological potential of the plants in the context of antiviral activity and justify the selection of the plants. Or it should be moved to Supplementary material

Response:   Table 1 has been moved to the Introduction of the manuscript.

  • Remark: Line 143: correct: “rotationto form”

Response:   The text is corrected.

  • Remark: Figure 1 legend is misleading. See major comment regarding gallic acid and quercetin. Explanation for “DOPC” should be added.

Response:   Figure 1 is removed.

  • Remark: 2.2. section: This section needs to be rewritten taking into account my comment regarding quantification of gallic acid and quercetin. Moreover, reference 57 is incorrect. In mentioned paper Authors did not use the spectrophotometric method.

Response:  Section 2.2.2. is rewritten.

  • Remark: Line 213: “the test compounds” should be replaced by “ the tested extracts”. The same for line 263, 388 – please check all text.

Response:   We thank the esteemed reviewer for the remark. The corresponding correction was made and the whole manuscript was reviewed for other similar expressions.

  • Remark: 2 section title. Title should be corrected to: “Determination of the Amount of Encapsulated Extracts”. This section needs to be rewritten taking into account my major suggestion (also Table 4, and 3.3. section).

Response:  Section 3.2. is rewritten and the title is corrected. There are corrections in Table 4.  

  • Remark: “the total amount of TPC and TFC is ca. 5 %” – TPC and TFC is not express as a % but as an equivalent of gallic acid and quercetin, respectively. Lines 333-336 are unnecessary and should be removed.

Response:   Lines 333-336 are removed.

  • Remark: Figures 3 and 4 are unnecessary and should be removed.

Response:   Figures 3 and 4 are removed.

  • Remark: Table 6: Why was a specific value not provided for unloaded liposomes?

Response:   Several types of unloaded liposomes were investigated in our previous study. They showed no cytotoxicity nor antiviral activity even at 3000 µg/mL. For the present study, they were prepared at a concentration of 2000 µg/mL. In order for the cell line to survive the treatment period, the stock suspension had to be diluted 1:1 with the nutrient medium required by the cells (this dilution proved to be sufficient for cell survival). Thus, the highest concentration of unloaded liposomes with which the cells were treated in the present experiment was 1000 µg/mL. Since no toxicity has been reported with it, we state that CC50 ≥ 1000 µg/mL.

  • Remark: The Discussion section should be rewritten. Some parts  of the text describe the obtained results. The results related to cytotoxicity are not discussed. Section 4.2: Antiviral Activity: Lines 525-543 are not related to antiviral activity; they describe different types of drug delivery systems. This content should be moved to Section 4.1.

Response:   Suggestions from the respected reviewer passage from section 4.2. has been moved to section 4.1.

Round 2

Reviewer 1 Report

Comments and Suggestions for Authors

No further comments.

Comments on the Quality of English Language

None

Author Response

We thank the esteemed reviewer for the recommendations he offered to improve our manuscript. We also thank him for the high evaluation he gave to our research and the way we presented it.

Reviewer 3 Report

Comments and Suggestions for Authors

The manuscript has been revised, and the authors have provided a detailed response to the comments. However, I still have some suggestions:

Line 85: Add an additional statement that specifically describes what is presented in Table 1 (e.g., "Table 1 summarizes...").

Table 1: "Plant extracts" — the title of the table is too vague. Specify what Table 1 specifically shows.

Section 2.2.4: "Release of quercetin and gallic acid…" and Section 3.3: "Release of TFC and TPC from the liposomes" — Correct the section titles. In my opinion, an expression like "release of the extract based on TPC and TFC" (or something similar) would be more accurate. Please review the entire manuscript in this context.

Line 251: "The release of TFC and TPC..." - correct as indicated above

Table 4: "Determination of the encapsulation efficiency of TFC and TPC expressed in" — Correct to "based on TFC and TPC…".

Line 403: "Spectroscopyrevealed" — Correct the spacing.

Line 416: "(The extracts are mixtures of multiple constituents and not only of quercetin and gallic acid.)" — This sentence is unnecessary; consider removing it.

Author Response

Comments 1: Line 85: Add an additional statement that specifically describes what is presented in Table 1 (e.g., "Table 1 summarizes...").

Response 1: An explanation of what Table 1 represents has been added.

Comments 2: Table 1: "Plant extracts" — the title of the table is too vague. Specify what Table 1 specifically shows.

Response 2: The title of Table 1 has been corrected to present its contents in more detail.

Comments 3: Section 2.2.4: "Release of quercetin and gallic acid…" and Section 3.3: "Release of TFC and TPC from the liposomes" — Correct the section titles. In my opinion, an expression like "release of the extract based on TPC and TFC" (or something similar) would be more accurate. Please review the entire manuscript in this context.

Comments 4: Line 251: "The release of TFC and TPC..." - correct as indicated above

Response 3 and 4: We appreciate the suggested corrections. Section titles have been corrected. The manuscript was reviewed and the phrase "release of the extract based on TPC and TFC" was used where appropriate.

Comments 5: Table 4: "Determination of the encapsulation efficiency of TFC and TPC expressed in" — Correct to "based on TFC and TPC…".

Response 5: The suggested correction has been made.

Comments 6: Line 403: "Spectroscopyrevealed" — Correct the spacing.

Response 6: The suggested correction has been made.

Comments 7: Line 416: "(The extracts are mixtures of multiple constituents and not only of quercetin and gallic acid.)" — This sentence is unnecessary; consider removing it.

Response 7: The sentence has been removed.